# VINOGROUND: TODAY'S LMMS DON'T UNDERSTAND SHORT COUNTERFACTUAL VIDEOS

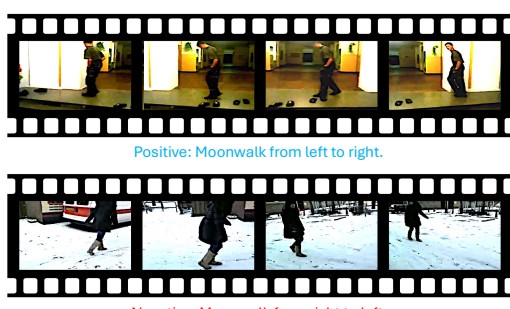
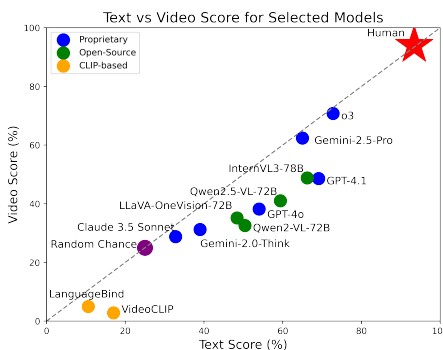

Figure 1: (Left) An example data point from the *spatial* category. Both videos last less than 10 seconds. Each data point contains two pairs of video-caption pairs counterfactual to each other. (Right) Models perform significantly poorer than humans, performing better on the text score metric than on the video score metric, as defined in Section 4.1.

## ABSTRACT

There has been growing sentiment recently that modern large multimodal models (LMMs) have addressed most of the key challenges related to short video comprehension. As a result, both academia and industry are gradually shifting their attention towards the more complex challenges posed by understanding long-form videos. However, is this really the case? Our studies indicate that LMMs still lack many fundamental reasoning capabilities even when dealing with short videos. We introduce Vinoground, a temporal counterfactual LMM evaluation benchmark encompassing 1000 short and natural video-caption pairs. We demonstrate that existing LMMs severely struggle to distinguish temporal differences between different actions and object transformations. For example, the best model o3 only obtains ∼50% on our group score metric, showing a large gap compared to the human baseline of ∼90%. All open-source multimodal models and CLIP-based models perform much worse, producing mostly random chance performance. Through this work, we shed light onto the fact that temporal reasoning in short videos is a problem yet to be fully solved. We will publicly share our benchmark.

## 1 INTRODUCTION

Large multimodal models (LMMs) have become very competitive in not only image comprehension but also short video comprehension. Proprietary models such as o3 (OpenAI, 2025a), GPT-4.1 (OpenAI, 2025b) and Gemini-2.5-Pro (Gemini Team, 2025) as well as open-source models like Qwen2.5-VL-72B (Bai et al., 2025) and InternVL3-78B (Zhu et al., 2025) demonstrate strong performance in summarizing a short video's contents and answering questions regarding its details. Recent SoTA reasoning models such as o3 and Gemini-2.5-Pro show powerful multimodal reasoning capabilities over images and videos alike. This has led many researchers to believe that short video comprehension has mostly been solved, and consequently, the community's focus has been increasingly trending toward creating models that understand longer-form videos that are 10s of seconds or even minutes long. Our study, however, indicates that existing models are far from being capable of fully understanding short videos that are just a few seconds long, especially when there is dense temporal information.

As demonstrated in Wu (2024) and Mangalam et al. (2023), for many existing video benchmarks like EgoSchema (Mangalam et al., 2023), ActivityNet-QA (Yu et al., 2019), MSVD and MSRVTT (Xu et al., 2017), the performance of most modern LMMs does not vary significantly with number of sampled frames. In fact, it is often the case that an LMM only needs to see a single frame to produce a correct response. This 'single-frame bias' (Lei et al., 2023) reduces the video comprehension problem into the much easier image comprehension problem, essentially discarding the temporal aspect of a video. Researchers have also proposed harder temporal counterfactual benchmarks (Li et al., 2024c; Wang et al., 2023; Mangalam et al., 2023; Saravanan et al., 2024; Liu et al., 2024b; Shangguan et al., 2025; Hong et al., 2025) in order to better evaluate an LMM's temporal understanding capabilities. Existing counterfactual datasets test a model's ability to distinguish slight changes from a video's original (positive) caption to the new (negative) caption by asking the model to match the video with the correct caption. However, they either do not contain any negative videos corresponding to the negative caption, or simply swap the order of two unrelated videos to form the positive and negative videos, making it easy to distinguish the negative pair from the original positive pair due to the videos' unnaturalness. Hence, these benchmarks may be inflating the performances of modern LMMs in understanding short videos.

In this paper, we introduce Vinoground, a temporal counterfactual LMM evaluation benchmark composed of 1000 short and natural video-caption pairs. Vinoground is a challenging benchmark aimed to expose the incapabilities of state-of-the-art models in understanding temporal differences between different actions (e.g., "the man eats then watches TV" vs. "the man watches TV then eats") and object transformations (e.g., "water turning into ice" vs. "ice turning into water"). In each pair of captions, the positive and negative are the same in word composition but different in order. Our work is inspired by Winoground (Thrush et al., 2022), a challenging counterfactual benchmark for visio-linguistic compositional reasoning in images. In Winoground, a model must correctly match two images with their corresponding captions, where both captions use the same set of words, but are rearranged to describe each image (e.g., "some plants surrounding a lightbulb" vs. "a lightbulb surrounding some plants"). This evaluates whether a model effectively encodes the text and images, paying attention to their compositional structures, and whether it can integrate and synthesize information across both modalities. Our benchmark's name changes the 'W' to a 'V' for "video", and further employs temporal counterfactuals to emphasize this unique element in video data. We use text score, video score, and group score to evaluate a model's ability to choose the right caption for a video, to choose the right video for a caption, and to match both positive and negative video-caption pairs correctly, respectively. These measure a model's textual, visual, and temporal reasoning capabilities in a balanced manner. Most of our videos are less than 10 seconds long, yet we find a very large performance gap between an average human and today's best models. An example can be found in Figure 1. We purposely focus on short videos as they efficiently expose deficiencies in temporal reasoning without the cost of long video curation and evaluation. Additionally, they prevent failures from being misattributed to limited context windows to process long videos rather than poor temporal understanding. If LMMs cannot handle short videos, tackling long ones is futile.

In sum, our main findings and contributions are:

- Existing benchmarks fail to fully expose LMMs' incapability in temporal reasoning.
- We introduce Vinoground, the first temporal and natural counterfactual evaluation benchmark for evaluating video understanding models using only short videos.
- Modern SoTA LMM performance is subpar when it comes to temporal reasoning in short video comprehension tasks; most models perform at random-chance level on video score and even worse on group score, both being significantly lower than text score.
- We categorize our data into 3 major categories, 'object', 'action', and 'viewpoint', as well as 4 minor categories, 'interaction', 'cyclical', 'spatial', and 'contextual', in order to dissect each model's capabilities for each of these categories. We find that existing models are decent at analyzing video frames at coarse-level but tend to miss fine-grained details.
- Short video comprehension is a problem that is far from being solved.
- Finally, Vinoground is comparable in scale to previous influential benchmark-only efforts like Winoground (Thrush et al., 2022) but demonstrably more challenging. Our benchmark is specifically designed for evaluation, and thus does not include training data. We believe Vinoground will provide a solid foundation for future research to develop and test better models facilitated by Vinoground's design philosophy and methodological rigor as demonstrated in Appendix N.

## 2 RELATED WORK

**Counterfactual Reasoning.** Counterfactual reasoning (Morgan & Winship, 2015) in the context of computer vision typically involves curating negative images and captions by manipulating the original data and observing how the outcome changes (Hendricks et al., 2018; Yeh et al., 2019; Goyal et al., 2019; Verma et al., 2020; Guo et al., 2023; Zhang et al., 2021; Thrush et al., 2022; Le et al., 2023; Zhang et al., 2024a). The idea is that a model should understand cause and effect and be able to make predictions in unseen situations. For evaluation, curating meaningful and hard negatives is important. Winoground (Thrush et al., 2022) is a pioneering benchmark for counterfactual reasoning where each data point contains two images and two corresponding captions. Given an image, a vision-language model is asked to find the matching caption from the provided two options, and vice versa. COCO-Counterfactual (Le et al., 2023) explores simple linguistic rules to generate negative captions and uses an image editing model to produce negative images. We introduce a novel benchmark with counterfactuals that are temporal, an attribute specific to the video modality.

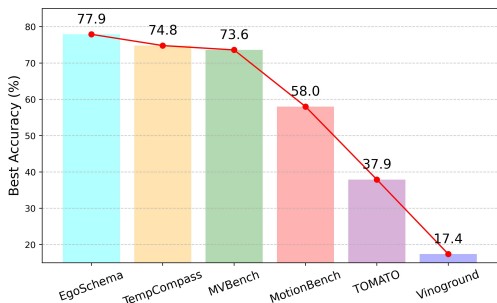

Figure 2: The performances of Qwen2-VL-72B on Vinoground and other temporal benchmarks. We can see that Vinoground is the most challenging benchmark.

**Single-Frame Bias and Temporal Reasoning.** An important aspect of video data is its temporality, i.e., how events change as time progresses. Modern LMMs sample frames and treat the video as a set of images, both during training and evaluation. Benchmarks such as EgoSchema (Mangalam et al., 2023), MSVD and MSRVTT (Xu et al., 2017) exhibit a 'single-frame bias' (Lei et al., 2023) where only one video frame is needed for a model to predict correctly, as a model's performance does not vary significantly as the number of frames sampled increases (Wu, 2024; Mangalam et al., 2023). To better evaluate a model's temporal understanding capabilities, researchers have developed datasets such as YouCook2 (Zhou et al., 2018), ActivityNet-QA (Yu et al., 2019) and COIN (Lin et al., 2022), which mainly involve procedural activities that often have a specific temporal dependency (e.g., if a video shows a person washing and slicing apples, and then baking an apple pie, a model would easily predict that "bake it to make a pie before washing the apple" is a wrong caption even without looking at the video). In contrast, Vinoground also includes actions that are entirely unrelated, such as "people are talking before drinking" vs "people are drinking before talking", making it more challenging for models to infer answers based solely on textual cues. MVBench (Li et al., 2024b) also includes temporal data that involves 20 different subcategories of temporal reasoning. However, even with this coverage, it does not contain any negatives like ours, reducing their difficulty since they do not contain any counterfactual examples. Beyond the absence of negative videos, NExT-QA (Xiao et al., 2021), ACQUIRED (Wu et al., 2023), and Causal-VidQA (Li et al., 2022) include temporally rich questions but mixes event inference (e.g. "what-if" questions) with temporal reasoning. In contrast, Vinoground isolates pure temporal reasoning by presenting events e.g., A and B explicitly and asking about their order—removing confounding factors like causality or inference.

**Temporal Counterfactuals.** Recent benchmarks combine counterfactuals with temporal reasoning. EgoSchema (Mangalam et al., 2023) introduces long-form videos where each video has 1 positive caption and 4 negative captions to choose from, while VITATECS (Li et al., 2024c) introduces temporal counterfactual data where a word or phrase is swapped/replaced from the positive caption to form the negative caption. However, neither has any negative videos and thus do not fully evaluate an LMM's dense temporal reasoning capabilities like we do. VELOCITI (Saravanan

Table 1: Comparison between Vinoground and other temporal datasets. Ours is the only one possessing natural negative videos that are counterfactual and mostly shorter than 10s.

| Dataset | Negative Videos | Counterfactual | Short (Avg ≤10s) | Natural Videos |
|---|---|---|---|---|
| Paxion | ✓ | ✓ | ✗ | ✗ |
| NExT-QA | ✗ | ✓ | ✗ | ✓ |
| MVBench | ✗ | ✗ | ✗ | ✓ |
| EgoSchema | ✗ | ✓ | ✗ | ✓ |
| VITATECS | ✗ | ✓ | ✗ | ✓ |
| VELOCITI | ✓ | ✗ | ✓ | ✓ |
| TempCompass | ✓ | ✓ | ✗ | ✗ |
| TOMATO | ✗ | ✓ | ✓ | ✗ |
| MotionBench | ✗ | ✗ | ✓ | ✓ |
| **Vinoground (Ours)** | ✓ | ✓ | ✓ | ✓ |

et al., 2024) introduces positive/negative videos as a part of their intra-video association benchmark by clipping random portions in the same video, and asking the model to distinguish between the events. These videos, however, are not truly counterfactual pairs as different clips within the same movie are not guaranteed to have a positive-negative relation. TempCompass (Liu et al., 2024b) includes videos that tests a model's ability to differentiate the order of events, but the videos are either concatenations of two completely unrelated videos with drastic frame changes in between the events, or reversed in time and thus impossible to happen in real life, and do not belong to the true data distribution. LMMs tend to do much better when it comes to such videos when compared to our benchmark's more natural negative videos, as shown in Fig. 2.

Similar to TempCompass, Paxion (Wang et al., 2023) uses reversed videos and caption edits (e.g., word swaps), which are often synthetically unnatural and detectable by models. Also, Paxion's perturbations are limited to captions, whereas Vinoground includes true negative videos, further increasing task difficulty. TOMATO (Shangguan et al., 2025) also partially incorporates generated/edited videos that are unnatural and do not have any negative videos. MotionBench (Hong et al., 2025), similarly, only uses positive videos. We summarize the comparisons between other temporal benchmarks and Vinoground in Table 1, demonstrating how Vinoground is the only benchmark unifying the four qualities, making it the most novel temporal reasoning benchmark. We provide further explanation of how Vinoground eliminates the confounder of static, atemporal reasoning in Appendix N.1.

## 3 VINOGROUND

In this section, we introduce our data curation and categorization process. In order to curate Vinoground's video-caption pairs, we first explain how we generate the required captions in Section 3.1, how we find the corresponding videos in Section 3.2, and finally the details of categorizing the videos in Section 3.3. An illustration of the overall process can be found in Appendix A.

### 3.1 GENERATING COUNTERFACTUAL CAPTIONS

The first step in curating our data is to find counterfactual caption pairs. We want to ensure that the captions we curate are of high-quality and temporal in nature. While human annotation is a possible solution, it is costly and difficult to scale up. Instead, we leverage a SoTA LLM, specifically the GPT-4 (OpenAI, 2024b) model, as it is much cheaper, follows the multiple requirements we impose, and guarantees that there are no duplicate candidates. We require our caption pairs to be composed of the exact same words, only permuted into different orders. We also want to avoid candidates that could easily be solved by looking at a single frame of the video such as "a man is waving at a woman" vs. "a woman is waving at a man". Hence, we ask GPT-4 to create *temporal* counterfactuals that require one to process and understand the entire video, and in particular, understand the order of events in which they happen, such as "a man waves at a woman before he talks to her" vs. "a man talks to a woman before he waves at her". We will later showcase in Section 4.3 that we can already expose LMMs greatly with such videos (i.e., by swapping the order of two events), making more complicated scenarios unnecessary. We include the detailed prompt fed to GPT-4 in Appendix E.

### 3.2 VIDEO CURATION

After curating counterfactual caption candidates, we gather corresponding videos for those captions. We utilize the VATEX (Wang et al., 2019) dataset, which contains 5 distinct captions for each maximum 10-second long video. We only use the val and test subsets of VATEX to make sure none of Vinoground is ever used as training data. This results in a pool of 9000 videos and 45000 captions.

We retrieve potential matches in VATEX according to the generated caption candidates. We leverage sentence transformers (Song et al., 2020), which are good at summarizing sentence-level information into feature vectors, to extract the features of both our GPT-generated captions and VATEX's captions. We subsequently use the Faiss library (Douze et al., 2024) to efficiently index and retrieve the top 20 most similar VATEX captions for each GPT-4 generated caption. We manually examine if any retrieved caption is a good match, and if its corresponding video reflects the caption as well. The primary criterion during manual review is straightforward: Does the caption accurately and unambiguously describe the video content? While this process does involve some degree of semantic judgment—as is inevitable in aligning language and vision—we mitigate subjectivity by (1) cross-validating questionable cases, and (2) filtering out ambiguous matches. We also ensure that only

caption/video pairs where multiple authors independently agree are retained. The quality of the dataset yielded under this process can be justified by our human performance (Table 3). For some cases where none of the retrieved captions are a good match, we search YouTube with the caption candidate to find a matching video.

In the end, we curate 500 counterfactual pairs of video-caption pairs (1000 video-caption pairs in total) for evaluation. Each video-caption pair is provided in the form of the original YouTube ID, the clip's starting and ending timestamps, and the corresponding caption. We also put Vinoground through 3 rounds of human evaluation by the authors, making sure that the pair of captions truly contain the same word composition and that the video clips indeed reflect their respective captions.

## 3.3 CATEGORIZATION

Table 2: The number of data points in Vinoground assigned under each category, separated by major and minor groups. All 500 pairs have one and only one major category assigned to them, while minor category assignments are content-based.

| **Major** | Object | Action | Viewpoint | Total |
|---|---|---|---|---|
| Count | 160 | 257 | 83 | 500 |
| **Minor** | Interaction | Cyclical | Spatial | Contextual |
| Count | 73 | 111 | 103 | 63 |

Finally, we want to be able to evaluate LMMs in a fine-grained manner on multiple aspects represented by our dataset. Hence, we categorize Vinoground according to the unique characteristics discovered through the data curation process. We report the number of counterfactual data pairs assigned under each category in Table 2. We define each category as follows.

We divide Vinoground into 3 major categories: *object*, *action*, and *viewpoint*. Each counterfactual pair must be in one and only one of the three major categories.

- **Object** requires LMMs to detect changes in the status of one specific object, such as "water turning into ice" vs. "ice turning into water." This category is similar to the "Reversing" category in TempCompass (Liu et al., 2024b) that evaluates a model's ability to detect attribute and directional changes. While TempCompass reverses positive videos in time to create negatives and thus can be unnatural, we curate real, natural videos that correspond to the negative captions.
- **Action**, on the other hand, simply asks models to distinguish the order in which two or more different actions happened, e.g. "the man eats and then watches TV" vs. "the man watches TV and then eats." The two actions need not be correlated at all, and thus less logical comprehension is necessary for a correct prediction.
- **Viewpoint** specifically describes changes in the camera angle, perspective, or focus within the video, such as "a person films the car in front of him before he films himself" vs. "a person films himself before he films the car in front of him." The change in viewpoint is usually accompanied by a drastic difference in between the frames, whereas other events most likely happen within the same context or background.

We also introduce 4 minor categories: *interaction*, *cyclical*, *spatial*, and *contextual*. Some pairs belong to a multitude of these minor categories, while some do not belong to any.

- **Interaction** involves videos where a human changes their way of interacting with an object in the course of the video, e.g. "the calligrapher writes with his pen before he dips it into the ink" vs. "the calligrapher dips his pen into the ink before he writes with it."
- **Cyclical** tests a model's ability to identify either procedural temporal activities or two actions that are dependent on each other. The calligrapher example earlier is also cyclical as the person repeats the procedure "write, dip, write, dip...", and the action "dip" happens as a result of "write" in the positive, while "write" is enabled after "dip" in the negative. In contrast, the "action" category can involve completely unrelated actions.
- **Spatial** It has been shown that LMMs struggle to distinguish physical locations between objects in image-caption pairs (Zhang et al., 2024a). We want to further evaluate this deficiency when it comes to temporal understanding as well. Thus, this category involves object movements and requires positional understanding, such as "the man ran from left to right" vs. "the man ran from right to left." Note that this does not include movement of the background; e.g., when the camera is moving along with the object in question, which belongs to the next category.
- **Contextual** requires LMMs to understand changes in the background or general information of entire video frames. An example is the pair "the biker rides down the street before he goes down the stairs" vs. "the biker goes down the stairs before he rides down the street" where the camera that

records the videos is strapped on the biker's forehead, making the background the only changing aspect. One cannot infer positional changes only by observing object movements like the "spatial" category, but instead must focus on the background as the object in question can appear motionless due to the camera moving along with the object.

We provide in-depth analysis of models' performances on our benchmark based on the above categories in Section 4.4.2. A detailed teaser can be found in Appendix M.

# 4 EXPERIMENTS

In this section, we evaluate state-of-the-art vision-language models on our benchmark. We first describe the models and evaluation metrics in Section 4.1; then we explain our experimental setup, including prompting methods and human studies, in Section 4.2; we analyze the performances of the models in Section 4.3, and provide further ablation studies in Section 4.4.

## 4.1 MODELS AND EVALUATION METRICS

We evaluate both CLIP-based (Radford et al., 2021) and large generative models, both proprietary and open-source. The exact list of models we evaluate can be found in Table 3. CLIP-based models use contrastive learning between videos and captions, while text-generation LMM models use next-token prediction to generate a response. Due to the different nature of the CLIP-based vs. LMM methods, we introduce our metrics in different fashions accordingly.

We use $C$ to denote captions and $V$ to denote videos. For each positive and negative set of counterfactual video-caption pairs, $(C_i, V_i)$ and $(C_i', V_i')$, $\forall i \in \{1, 2, ..., 500\}$, we ask CLIP-based models to compute a similarity score $e$ between not only the correct pairs but also the incorrect pairs $(C_i, V_i')$ and $(C_i', V_i)$ (identical to Winoground (Thrush et al., 2022)). For generative LMMs, we can only provide inputs (e.g., 2 captions and 1 video) to the model and ask it to choose between the captions/videos.

We first evaluate the text score $s_t$ where the model is presented with both positive and negative captions but only one of the videos, forming the triplets $(C_i, C_i', V_i)$ and $(C_i, C_i', V_i')$. For each triplet, the model is then asked to choose the caption that describes the contained video. We denote the score function of a model response given any triplet as $s$; for instance,

$$s(C_i, C_i', V_i) = \begin{cases} 1 & \text{if LMM chooses } C_i \text{ or} \\ & e_{(C_i, V_i)} > e_{(C_i', V_i)} \text{ for CLIP-based} \\ 0 & \text{otherwise} \end{cases} \qquad s(C_i, C_i', V_i') = \begin{cases} 1 & \text{if LMM chooses } C_i' \text{ or} \\ & e_{(C_i', V_i')} > e_{(C_i, V_i')} \text{ for CLIP-based} \\ 0 & \text{otherwise} \end{cases}$$

Then the text score for the given counterfactual pair $(C_i, V_i)$ and $(C_i', V_i')$ is:

$$s_t(C_i, C_i', V_i, V_i') = s(C_i, C_i', V_i) \wedge s(C_i, C_i', V_i')$$

where $\wedge$ is the logical and operator; i.e., $s_t$ is 1 only if both triplets are correct. This exposes the models when they guess randomly.

Similarly, for video score $s_v$, the model is presented with one caption and both positive and negative videos, forming triplets $(C_i, V_i, V_i')$ and $(C_i', V_i, V_i')$. For each triplet, the model is asked to choose the video that is described by the caption. In this case, the response scoring becomes:

$$s(C_i, V_i, V_i') = \begin{cases} 1 & \text{if LMM chooses } V_i \text{ or} \\ & e_{(C_i, V_i)} > e_{(C_i, V_i')} \text{ for CLIP-based} \\ 0 & \text{otherwise} \end{cases} \qquad s(C_i', V_i, V_i') = \begin{cases} 1 & \text{if LMM chooses } V_i' \text{ or} \\ & e_{(C_i', V_i')} > e_{(C_i', V_i)} \text{ for CLIP-based} \\ 0 & \text{otherwise} \end{cases}$$

Then the video score is:

$$s_v(C_i, C_i', V_i, V_i') = s(C_i, V_i, V_i') \wedge s(C_i', V_i, V_i')$$

We also include a group score metric $s_g$:

$$s_g(C_i, C_i', V_i, V_i') = s_t(C_i, C_i', V_i, V_i') \wedge s_v(C_i, C_i', V_i, V_i')$$

$s_g$ serves as the ultimate test for a model to demonstrate its temporal reasoning capabilities in both the textual and visual domains, as both $s_t$ and $s_v$ must be 1. For all three metrics, we report the mean over all test instances. We include an illustration of the metrics in Appendix B.

## 4.2 EXPERIMENTAL SETUP

Since for each pair of counterfactuals, we have 2 text-score questions and 2 video-score questions, we have 2000 questions in total. To evaluate CLIP-based models, we use the evaluation code provided by the authors to calculate video-caption embeddings and similarity scores. Evaluating text-generative models is slightly more complicated. We first introduce the different prompts we use. For text score, we provide the model with the video and the two corresponding captions, and prompt "⟨video⟩ Which caption best describes this video? A. {Caption 1}, B. {Caption 2}". For video score, however, since some LMMs only support 1 video input, we concatenate the positive and negative videos into a single video with a 2 second black screen in between, easily identifiable by evaluated models.

When sampling $N$ frames, we make sure we sample $(N-1)/2$ frames from the positive and negative video fragments and at least 1 frame of black screen in between. More details can be seen in Appendix L. For the sake of consistency, we provide all models with the concatenated video, regardless of how many videos they can actually take as input. We then prompt the model with "⟨video⟩ Which video segment matches this caption? Note: The video contains two segments separated by a 2-second black frame. Caption: {Caption}. A. First segment (before black frame), B. Second segment (after black frame)" to choose between the two video segments. For text score, we shuffle the caption orders so that both answer choices "A" and "B" have 50% probability as ground truths. For video score, we concatenate videos in random orders while also making sure both answer choices appear evenly. We also report the results with respect to the number of frames sampled by the model from the video, if supported, to evaluate the effect of temporality in Section 4.4.1. All experiments are done with 4xA100-80GBs.

In addition, we use Prolific (https://www.prolific.com) to evaluate human performance and find that our dataset is fairly easy for an average human to complete with high accuracy. Prolific is a platform similar to Amazon MTurk which recruits workers to complete tasks such as data annotation. The interface we present to the workers is in Appendix H. To filter out unfaithful workers, we first employ a qualification process by sampling 10 video-question pairs from TempCompass (Liu et al., 2024b) that are of the event order category, which contains concatenated videos with no correlation, such as "a man lifts weights in a gym, then a cat plays on the grass". Such examples are easy enough for an average human to obtain 100% accuracy. We ask the workers the 10 beginner-level questions first, and they are qualified only if they answer every question correctly. This process results in 170 qualified workers (demographics included in Appendix H).

Table 3: Results for different models and number of sampled frames. Performances significantly better than random chance are bolded. There are four groups separated by double lines: random chance and human performance, proprietary text-generative models, open-source text-generative models, and CLIP-based models from top to bottom. The best performances of proprietary and open-source models are highlighted in red.

| Model | # Fr | $s_t$ | $s_v$ | $s_g$ |
|---|---|---|---|---|
| Random Chance | N/A | 25.0 | 25.0 | 16.7 |
| Prolific Human | All | 93.4 | 94.0 | 90.0 |
|  | 32 | 91.4 | 90.8 | 85.2 |
| **Proprietary Large Multimodal Models** | | | | |
| o3 (OpenAI, 2025a) | 32 | 72.8 | 72.2 | 56.6 |
| GPT-4.1 (OpenAI, 2025b) | 32 | 69.1 | 48.6 | 38.5 |
| o4-mini (OpenAI, 2025a) | 32 | 58.2 | 40.9 | 28.5 |
| GPT-4o (CoT) | 32 | 59.2 | 51.0 | 35.0 |
| GPT-4o (OpenAI, 2024a) | 32 | 54.0 | 38.2 | 24.6 |
| GPT-4o | 0 | 10.0 | 24.6 | 2.0 |
| Gemini-2.5-Pro (Gemini Team, 2025) | 1fps | 65.0 | 62.4 | 49.0 |
| Gemini-2.5-Flash (Gemini Team, 2025) | 1fps | 60.0 | 54.2 | 41.8 |
| **Open-Source Large Multimodal Models** | | | | |
| InternVL3-78B (Zhu et al., 2025) | 32 | 66.2 | 48.8 | 36.2 |
| InternVL3-8B (Zhu et al., 2025) | 32 | 48.0 | 28.2 | 14.6 |
| Qwen2.5-VL-72B (Bai et al., 2025) | 32 | 59.4 | 41.0 | 29.6 |
| Qwen2.5VL-7B (Bai et al., 2025) | 32 | 47.6 | 34.4 | 19.2 |
| InternVL2.5-78B (Chen et al., 2025) | 32 | 58.0 | 40.6 | 27.6 |
| InternVL2.5-8B (Chen et al., 2025) | 32 | 46.6 | 28.2 | 14.4 |
| Qwen2-VL-72B (CoT) | 32 | 53.0 | 26.6 | 15.2 |
| Qwen2-VL-72B (Wang et al., 2024) | 32 | 50.4 | 32.6 | 17.4 |
| LLaVA-Video-72B (Zhang et al., 2024c) | 64 | 49.2 | 34.0 | 20.2 |
| LLaVA-Video-7B (Zhang et al., 2024c) | 64 | 42.4 | 30.0 | 17.0 |
| VideoLLaMA3 (Zhang et al., 2025) | 16 | 47.4 | 30.4 | 15.6 |
| Apollo-7B (Zohar et al., 2024) | 4 | 43.8 | 30.2 | 17.2 |
| VideoLLaMA2-72B (Cheng et al., 2024) | 8 | 36.2 | 21.6 | 8.4 |
| MiniCPM-2.6 (Yao et al., 2024) | 16 | 32.6 | 29.2 | 11.2 |
| Aria (Li et al., 2025) | 32 | 34.8 | 28.8 | 12.0 |
| InternLM-XC-2.5 (CoT) | 1fps | 30.8 | 28.4 | 9.0 |
| InternLM-XC-2.5 (Zhang et al., 2024b) | 1fps | 28.8 | 27.8 | 9.6 |
| Video-LLaVA-7B (Lin et al., 2024) | 8 | 24.8 | 25.8 | 6.6 |
| LLaVA-NeXT-34B (CoT) | 32 | 25.8 | 22.2 | 5.2 |
| LLaVA-NeXT-34B (Liu et al., 2024a) | 32 | 23.0 | 21.2 | 3.8 |
| **CLIP-based Models** | | | | |
| VideoCLIP (Xu et al., 2021) | 60 | 17.0 | 2.8 | 1.2 |
| LanguageBind (Zhu et al., 2024) | 8 | 10.6 | 5.0 | 1.2 |
| ImageBind (Girdhar et al., 2023) | 20 | 9.4 | 3.4 | 0.6 |

first, and they are qualified only if they answer every question correctly. This process results in 170 qualified workers (demographics included in Appendix H).

We conduct human evaluation under two settings. First, the Prolific workers are provided the full videos with audio. We want to create another environment where the workers see the same input as the models. Hence, we uniformly sample 32 frames from each video and concatenate them into a new 10-second video with no audio. The results for the two settings are also compared in Section 4.4.1.

Each question is answered by 10 unique workers. For the 10 answers from a single question, we calculate the *average* human response by taking the mode of the 10 answers. We then report the mean over all the questions as the final result.

## 4.3 MAIN RESULTS

Table 3 presents the results. (Appendix K presents more detailed results, as we only include each model's best performances here.) First, all CLIP-based models (VideoCLIP, LanguageBind, ImageBind) perform much worse than random chance, suggesting that contrastive learning does not provide models with enough knowledge of temporality. Among text-generative models, o3 performs best, achieving 56.6% on the group score metric. Reasoning models or the use of Chain-of-Thought (CoT) prompting (Wei et al., 2022) further outperforms the performance of non-reasoning models, especially for GPT-4o where on the video score metric it improves by 12.8% and 10.4% on group score. We include the full CoT prompt and parsing process in Appendix F. Amongst the open-source models, InternVL3-78B performs best, matching the scores of closed-source model GPT-4.1, achieving third best amongst all evaluated models. Using CoT on open-source models, however, helps much less, especially if they perform at near chance level. Given the fact that many models perform at or worse than random chance, and even the best model has a 40% gap with humans on the group score metric, it is evident that dense temporal reasoning is still very challenging for LMMs. We further provide failure case analysis for some of GPT-4o's responses in Appendix I.

Similar to Winoground (Thrush et al., 2022), we find that for models that perform better than chance level, their text score is significantly higher than video score, while group score is the lowest amongst all three. Fig. 1 also demonstrates this pattern, reflecting models' strengths at identifying textual rather than visual/temporal differences. For example, GPT-4o's video score (38.2%) is significantly lower compared to its text score (54.0%). Many open-source models only have non-random outcomes on the text score but equal or lower than random chance on video and group scores.

Human evaluators perform significantly better than any model, with scores around 90%. This indicates that Vinoground can be tackled relatively easily within human capacity. When the human evaluators are provided with 32-frame videos, the scores decrease by a few points, but are still much higher than those of any model. We emphasize yet again the huge gap between human group score and the best model o3's group score of 56.6%.

Finally, we report results for GPT-4o with 0 sampled frames as a control to measure text bias. For the text score, we hypothesize that the model will select the more likely caption in the absence of visual input, and for the video score, we hypothesize it will choose an answer at random—both of which are observed in practice. The below-chance text score of 10.0% suggests a systematic language bias in GPT-4o, where it tends to prefer one caption over the other (if it did so consistently, the score would drop to 0). Our balanced scoring method—computing both $s(C_i, C_i', V_i)$ and $s(C_i, C_i', V_i')$—prevents models from achieving high scores purely through such biases. This stands in contrast to benchmarks such as VITATECS (Li et al., 2024c) and EgoSchema (Mangalam et al., 2023), which lack negative videos and therefore allow models to succeed simply by exploiting caption likelihood.

Overall, even the strongest models show limited performance on dense temporal reasoning, despite being evaluated on short videos (under 10 seconds). This highlights that short video comprehension in LMMs remains far from human-level intelligence. We discuss further insights on model design and data utilization strategies in Appendix N.4.

## 4.4 IN-DEPTH ANALYSIS OF PERFORMANCE VARIATIONS

### 4.4.1 FRAMES SAMPLED

We evaluate Vinoground's temporal understanding requirements by varying the number of frames sampled. If a dataset suffers from *single-frame bias*, model performance should remain largely unchanged when comparing 1 frame to multiple frames. However, as shown in Table 4 (with additional results in Appendix K), performance consistently improves with more frames, indicating that full video context is necessary to solve the task.

Interestingly, oversampling can degrade performance: for GPT-4o, the 64-frame variant performs 5% worse across all three metrics compared to the 32-frame variant. We hypothesize that current models struggle to filter redundant information and to separate signal from noise when overloaded with visual tokens. Appendix G provides further analysis in comparison to prior work, highlighting key differences in temporal behavior and model limitations.

Note that for our video score metric to function as intended, a model must sample at least one frame from each video, and at least one black frame in between. This means that the number of frames sampled must be no fewer than 3. We hence gray out the video score and group score performances of models sampled at 1 or 2 frames and only focus on their text scores.

Finally, for human evaluators, the 'All' group performs better than the 32 frame group, which indicates that humans can answer Vinoground questions better when the full videos are shown. In contrast, modern LMMs generally lack the ability to process inputs of an entire video without coarse sampling of frames. This suggests that further research into creating models that can handle more frames will be an important research direction for temporal reasoning.

Table 4: Performance of selected models with different # of frames sampled. Those significantly higher than random chance are highlighted, with the best performance of each model highlighted in red. Having more frames can improve performance, yet too many frames can also worsen it.

| Model | # Fr | $s_t$ | $s_v$ | $s_g$ |
|---|---|---|---|---|
| Prolific Human | All | **93.4** | **94.0** | **90.0** |
| | 32 | **91.4** | **90.8** | **85.2** |
| GPT-4o | 64 | **49.0** | **34.8** | 19.0 |
| | 32 | **54.0** | **38.2** | **24.6** |
| | 8 | **53.6** | **31.4** | 20.6 |
| | 1 | 28.2 | 28.0 | 10.0 |
| LLaVA-OneVision-72B | 64 | **46.2** | **31.8** | 18.6 |
| | 32 | **48.4** | **35.2** | **21.8** |
| | 16 | **47.2** | **33.8** | 20.4 |
| | 8 | **46.8** | **29.8** | 19.0 |
| | 4 | **40.4** | 24.8 | 13.0 |
| | 2 | **33.4** | 25.2 | 10.2 |

### 4.4.2 CATEGORY

Fig. 3 shows results per category as defined in Section 3.3. Interestingly, many models perform significantly better on the *viewpoint* and *contextual* categories, while being significantly worse on other categories. Here, we only report the group score for a selected set of models due to space. See Appendix J for the full results.

Both *viewpoint* and *contextual* bring forth drastic changes in between the video frames whenever the events

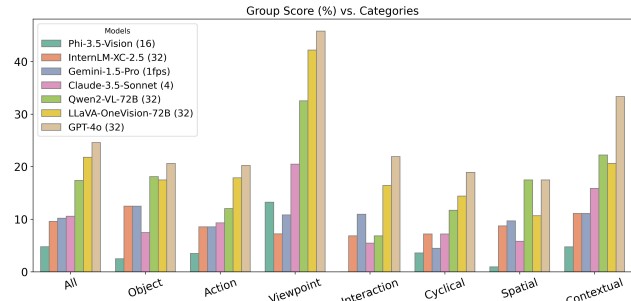

Figure 3: Group score of selected models per category. Models do better on contextual & viewpoint, worse on others.

change, as *contextual* involves background changes that occupy most of the frame while in *viewpoint*, as the camera angle changes, the entirety of the video frame changes as well. On the other hand, *interaction* and *cyclical* not only require a model to have strong logical understanding of the connection between events, but also the ability to focus on small temporal changes for the different actions involved. *Spatial*, as previously hypothesized, also poses a difficult challenge for models in understanding changes in object location. Overall, today's models are much better at understanding coarse-level information over a set of frames in their entirety than understanding fine-grained details from a part of each video frame. This also demonstrates how fine-grained comprehension is also crucial for dense temporal reasoning. We further explain the novelty of this finding in Appendix N.3.

## 5 CONCLUSION

We introduced Vinoground, a novel temporal counterfactual benchmark encompassing 1000 short and natural video-caption pairs. We demonstrated that existing video LMMs are quite incapable in terms of temporal reasoning, even for short (<10s) videos. While an average human can easily and accurately complete our benchmark, the best model, o3, performs much worse, and most models barely perform better than random chance. Our work demonstrates that there is much more to do still in the area of short video comprehension. We believe Vinoground can serve as an important checkpoint in evaluating a model's true performance for temporal understanding of short videos.

**Limitations.** One cannot fully analyze the behavior of proprietary models included in this paper due to the lack of access to their raw weights, such as o3, GPT-4.1, and the Gemini series.

**Reproducibility Statement.** We attach the dataset in the submission's supplementary materials. We will also publicly release it with the evaluation code upon the paper's acceptance.

**Ethics Statement.** All videos are sourced from publicly available academic datasets (VATEX) and platforms (YouTube). We will make the dataset and our code publicly available to ensure transparency.

**LLM Usage.** Gemini-2.5-Pro is used to polish only an insignificant portion of the paper's writing.

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

## APPENDIX

## A  DATA CURATION PROCESS

We include an overall illustration of the data curation process in Figure 4.

## B  METRICS ILLUSTRATION

We visualize our text and video score metrics in Figure 5. This shows the 4 possible questions that can be derived from one counterfactual data point in the dataset.

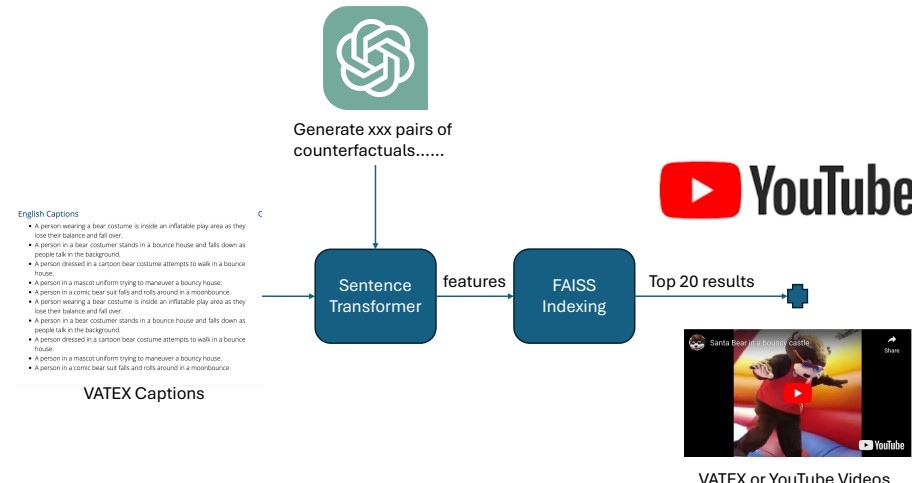

Figure 4: The data curation process.

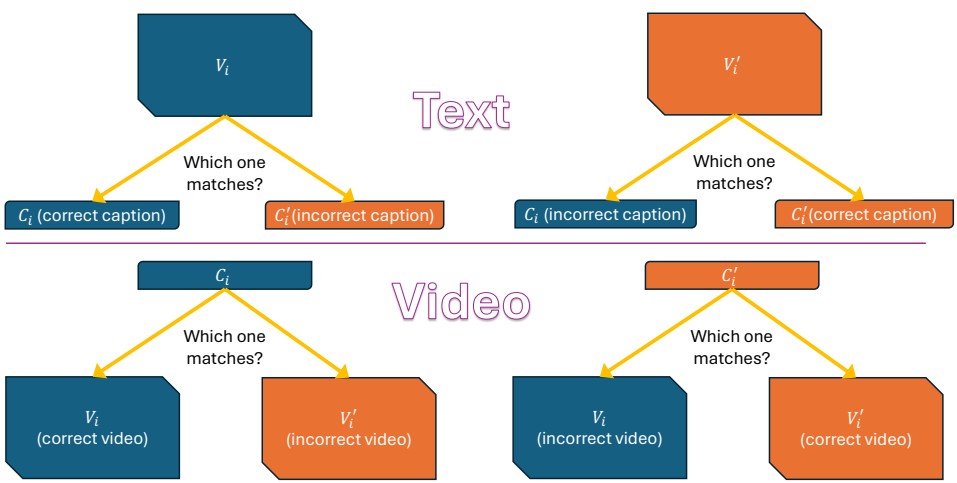

Figure 5: Visualization of the text and video score metrics.

## C  RANDOM CHANCE PERFORMANCE

We set the random chance performance for text, video, and group score as 25%, 25%, and 16.67%. It is intuitive to understand the setup for both text and video score since there are two questions in the same counterfactual pair for each metric, and the probability of guessing correctly is 50% each. For the counterfactual pair $(C_i, C_i', V_i, V_i')$, a model can only produce six possible permutations of video-caption matchings: $\{(C_i, V_i), (C_i', V_i')\}, \{(C_i, V_i), (C_i, V_i')\}, \{(C_i, V_i), (C_i', V_i)\},$ $\{(C_i, V_i'), (C_i', V_i')\}, \{(C_i', V_i), (C_i', V_i')\},$ and $\{(C_i', V_i), (C_i, V_i')\}$. This is why the random chance performance for group score is $1/6 \approx 16.7\%$.

## D  HUMAN PERFORMANCE

The 90% group score is because of human error, not because of data quality issues. Upon carefully examining the error cases, we find no particular pattern or poor quality examples. On the other hand, Panko (2008) shows how humans have a 5% error rate even for the simple task of entering spreadsheets, which closely models the human text and video scores 93% and 94%. Since group score is a composite metric of both, the combined correctness is $0.95 \cdot 0.95 = 0.9025$, which matches our human group score performance. This confirms the high-quality of Vinoground.

## E    Caption Curation Prompt

The prompt we give GPT-4 to generate potential caption candidates is: "I am trying to find videos that have appropriate temporal counterfactuals; e.g., I want to find video pairs that can be described with the following captions: "a man eats then watches TV" vs "a man watches TV then eats"; "the old man is working hard before the young man is playing" vs "the young man is working hard before the old man is playing". Note that for both elements of the same pair, they use the exact same words. Give me 10 examples." Then in the same conversation, we prompt the model "give me 10 different ones" until we have 500 pairs of candidates.

## F    CoT Prompt and Parsing

For chain-of-thought prompting, we simply add "please think step by step" at the end of our questions (as mentioned in Section 4.2). We then use GPT-4 as the judge with the prompt: "Please parse the following model response into either A or B. If the model response is just A or B, then it denotes the model answer, just output it. The model response starts after ====, and end before ====):\n==== ⟨MODEL RESPONSE⟩ ====\nProvide output your answer as a single character (A or B): "

## G    Comparison with Prior and Concurrent Analyses on Impact of Number of Frames Sampled

Our work provides a more nuanced analysis of the relationship between temporal sampling and model performance, extending the findings of prior studies. For instance, while Lei et al. (2023) effectively demonstrates a single-frame bias, our analysis in Section 4.4.1 reveals a more complex relationship. We show that for each model, there is an optimal range for the number of sampled frames; performance degrades not only when too many frames are used, but also when too few are sampled.

Furthermore, we offer a complementary perspective to the work of Du et al. (2024). Their results primarily illustrate a performance increase with up to 128 frames. In contrast, our findings highlight significant performance degradation with as few as 32 or 64 frames for certain models. To cleanly isolate the impact of temporal sampling density, our analysis in Section 4.4.1 varies only the number of frames. This controlled approach allows us to more directly attribute performance changes to the sampling rate.

In summary, our analysis moves beyond identifying single-frame bias to reveal a more granular, model-dependent relationship between frame count and performance, offering a precise investigation into the effects of temporal sampling.

## H    Prolific Survey Interface and Worker Demographics

We first upload all the videos to Google Drive and embed them into our surveys using Qualtrics. The 2000 questions from Vinoground are split into 50 surveys, with each survey having 40 random questions. We then distribute our surveys on Prolific where we pay everyone who completed a survey $2, or $0.05 per question. The interface is illustrated in Figure 7.

Out of the 170 qualified workers, 91 were male and 79 were female; 13 were students, 119 were non-students, and 38 have no data/did not consent to provide information; 69 were full-time working, 22 were part-time working, 20 were not in paid work (e.g. homemaker, retired or disabled), 17 were unemployed, and 42 have no data/did not consent to provide information; all of the 170 workers were fluent in English; we report the histogram of workers' ages in Figure 8.

## I    Analysis of Selected Failure Cases of GPT-4o

Here we analyze 3 example failure cases of GPT-4o on our benchmark. One can find the corresponding video clips from the supplementary materials. Extracted frames for video 8_neg are shown in Figure 6.

Figure 6: For this video, GPT-4o summarizes the video's event in the exact opposite order of the video itself, indicating its lack of understanding temporal dynamics between the frames.

For video 8_neg, the model response is: "a box with a lid placed on top of it → The hand interacts with the lid → The hand lifts the lid off the box → The hand moves the lid to the side of the box → the lid placed to the side of the box. Based on this sequence, the man moved the lid from the top of the box to the side". In this response, the model summarized the video's event in the exact opposite order of the video itself, indicating its lack of understanding temporal dynamics between the frames.

For video 248_pos, the model's response is: "the man is seen using the machine from left to right → the man continues moving the machine towards the right → the man is seen moving the machine back towards the left. Based on this analysis, the man first uses the machine from left to right and then goes from right to left." In this analysis, the model also reported the order of events in their exact opposite, but note the fact that here the model specifically failed to understand spatial movements as well.

For video 361_neg, the model response is: "two scallops on a grill, with the one on the left being more prominently visible → the camera appears to shift focus slightly, but the scallop on the left remains the primary focus → the camera seems to zoom in or shift slightly, but it does not distinctly pan from one scallop to the other. Since the video does not show a clear pan from one scallop to the other, neither caption A nor B describes the video." In this response, the model ignores huge frame shifts that is easily identifiable by the human eye. This shows how models can fail even with coarse-grain large frame changes.

Watch this video, and choose the correct caption out of the two that matches the video.

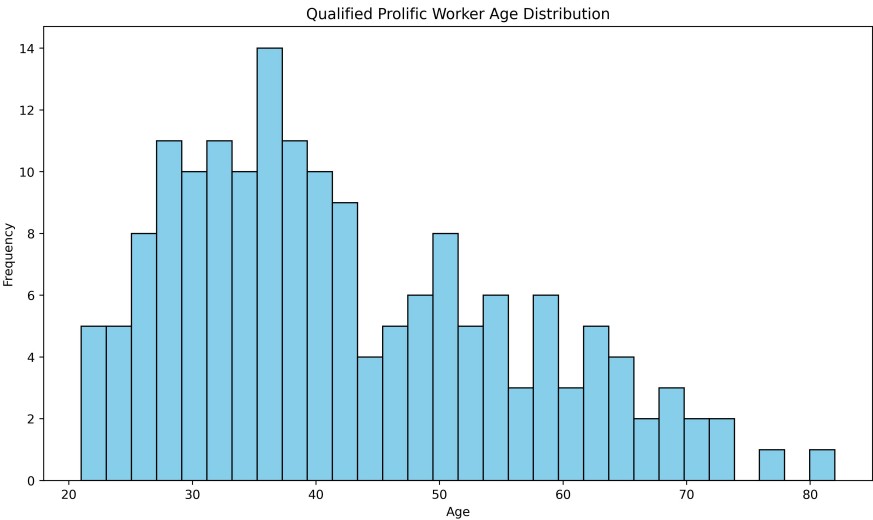

○ A. the person maneuvers other car parts before he shifts gears

○ B. the person shifts gears before he maneuvers other car parts

Figure 7: The Qualtrics survey that Prolific workers see.

Figure 8: Distribution of Prolific workers' ages.

## J    FULL CATEGORICAL RESULTS

Here we include the selected top-6 strongest models we evaluated and report their results by category in Tables 5 and 6. We also include the text and video score bar plots in Figures 9 and 10. We can see that the general trend is the same as reported in Section 4.4.2, where models perform much better on contextual and viewpoint, and worse on other categories.

Table 5: The best performances of proprietary models grouped by category. Significantly high performances are highlighted in blue, while significantly low performances are highlighted in red.

| category | GPT-4o | | | Gemini-1.5-Pro | | | Claude 3.5 Sonnet | | |
|---|---|---|---|---|---|---|---|---|---|
| | text | video | group | text | video | group | text | video | group |
| all | 54.00 | 38.20 | 24.60 | 35.80 | 22.60 | 10.20 | 32.80 | 28.80 | 10.60 |
| object | 52.50 | 35.62 | 20.62 | 36.25 | 25.62 | 12.50 | 30.00 | 25.00 | 7.50 |
| action | 47.47 | 35.41 | 20.23 | 30.74 | 22.18 | 8.56 | 27.63 | 28.79 | 9.34 |
| viewpoint | 77.11 | 51.81 | 45.78 | 50.60 | 18.07 | 10.84 | 54.22 | 36.14 | 20.48 |
| interaction | 50.68 | 42.47 | 21.92 | 30.14 | 27.40 | 10.96 | 20.55 | 21.92 | 5.48 |
| cyclical | 39.64 | 41.44 | 18.92 | 22.52 | 19.82 | 4.50 | 27.03 | 25.23 | 7.21 |
| spatial | 47.57 | 30.10 | 17.48 | 37.86 | 24.27 | 9.71 | 31.07 | 20.39 | 5.83 |
| contextual | 53.97 | 49.21 | 33.33 | 38.10 | 31.75 | 11.11 | 52.38 | 28.57 | 15.87 |

Table 6: The best performances of selected open-source models grouped by category. Significantly high performances are highlighted in blue, while significantly low performances are highlighted in red.

| category | LLaVA-OneVision-72B | | | Qwen2-VL-72B | | | InternLM-XC-2.5 | | |
|---|---|---|---|---|---|---|---|---|---|
| | text | video | group | text | video | group | text | video | group |
| all | 48.40 | 35.20 | 21.80 | 50.40 | 32.60 | 17.40 | 28.80 | 27.80 | 9.60 |
| object | 42.50 | 33.75 | 17.50 | 46.88 | 33.75 | 18.12 | 28.75 | 28.12 | 12.50 |
| action | 42.80 | 31.91 | 17.90 | 44.75 | 28.79 | 12.06 | 25.68 | 29.96 | 8.56 |
| viewpoint | 77.11 | 48.19 | 42.17 | 74.70 | 42.17 | 32.53 | 38.55 | 20.48 | 7.23 |
| interaction | 36.99 | 36.99 | 16.44 | 34.25 | 31.51 | 6.85 | 23.29 | 36.99 | 6.85 |
| cyclical | 36.04 | 29.73 | 14.41 | 36.94 | 32.43 | 11.71 | 18.92 | 36.04 | 7.21 |
| spatial | 37.86 | 25.24 | 10.68 | 53.40 | 31.07 | 17.48 | 23.30 | 29.13 | 8.74 |
| contextual | 57.14 | 31.75 | 20.63 | 49.21 | 39.68 | 22.22 | 26.98 | 26.98 | 11.11 |

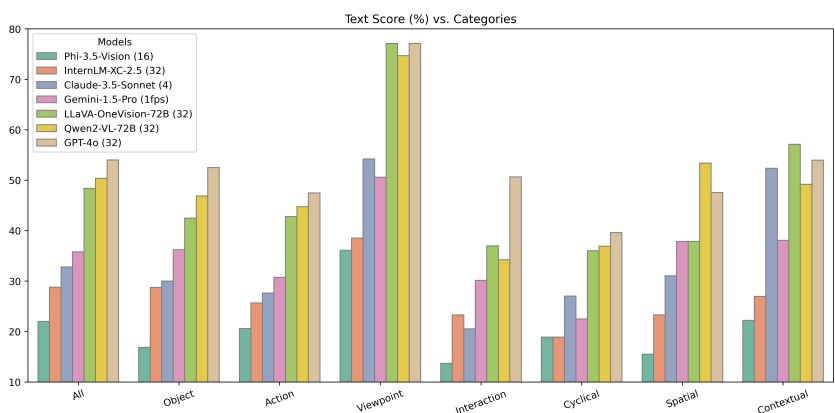

Figure 9: Text score bar plot based on category grouped by model.

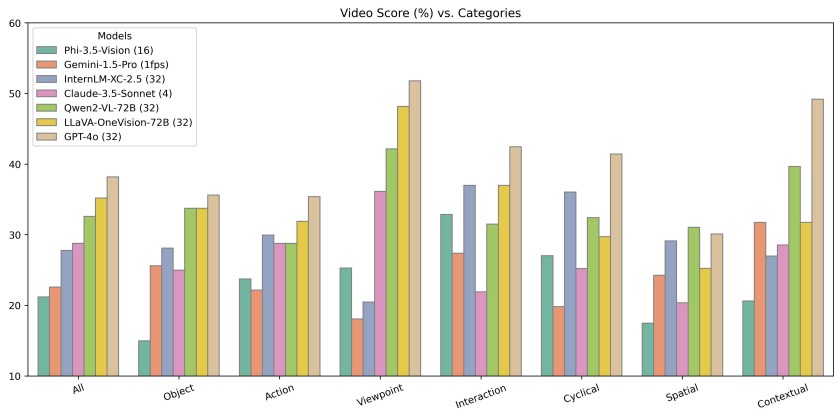

Figure 10: Video score bar plot based on category grouped by model.

## K FULL RESULTS ON EVALUATED MODELS

Due to the extensive number of models evaluated and different number of frames sampled as hyperparameters, we include the full results of our evaluation that are not mentioned in the main paper in Tables 7 and 8.

Table 7: The full evaluation results based on model type, frames sampled, and the metrics aforementioned. Only the model settings that are not mentioned in the main paper are listed here. Performances significantly better than random chance are bolded.

| Model | Frames | Text | Video | Group |
|---|---|---|---|---|
| OpenAI o1 OpenAI (2024c) | 32 | **59.1** | **50.5** | **36.0** |
| Gemini-2.0-Thinking (Google, 2025) | 32 | **39.0** | **31.2** | 14.6 |
| Gemini-1.5-Pro (CoT) | 1fps | **37.0** | 27.6 | 12.4 |
| Gemini-1.5-Pro (Gemini Team, 2024) | 1fps | **35.8** | 22.6 | 10.2 |
| Claude 3.5 Sonnet (CoT) | 4 | **39.4** | 27.0 | 13.6 |
| Claude-3.5-Sonnet | 16 | **30.0** | 22.6 | 8.4 |
| | 8 | **32.2** | 25.4 | 9.4 |
| | 4 | **32.8** | 28.8 | 10.6 |
| | 2 | 29.4 | 24.0 | 8.4 |
| | 1 | 26.2 | 30.0 | 10.8 |
| Qwen2-VL-72B | 32 | **50.4** | **32.6** | 17.4 |
| | 8 | **37.4** | 23.0 | 7.8 |
| | 4 | 26.2 | 23.8 | 6.2 |
| | 2 | 15.6 | 24.4 | 4.0 |
| Qwen2-VL-7B (Wang et al., 2024) | 32 | **40.0** | 26.4 | 11.8 |
| | 16 | **36.8** | 25.8 | 10.2 |
| | 8 | 27.6 | 23.4 | 7.8 |
| | 4 | 22.2 | 22.8 | 5.6 |
| | 2 | 21.4 | 25.8 | 5.2 |
| | 4fps | **40.2** | **32.4** | 15.2 |
| | 2fps | **34.8** | 27.4 | 10.6 |
| | 1fps | 26.8 | 26.6 | 7.6 |
| | 0.5fps | 23.2 | 19.6 | 4.8 |
| MiniCPM-2.6 | 32 | 28.4 | 27.0 | 9.4 |
| | 16 | **32.6** | **29.2** | 11.2 |
| | 8 | **33.4** | 25.6 | 9.0 |
| | 4 | 25.8 | 27.4 | 8.6 |
| | 2 | 22.8 | 23.2 | 4.6 |
| | 1 | 27.0 | 27.0 | 8.0 |
| LLaVA-NeXT-Video-34B (Liu et al., 2024a) | 32 | 23.0 | 21.2 | 3.8 |
| | 16 | 21.0 | 21.8 | 4.4 |
| | 8 | 21.2 | 22.0 | 5.2 |
| | 4 | 16.6 | 21.6 | 3.4 |
| | 2 | 15.4 | 21.6 | 2.2 |
| | 1 | 13.2 | 21.8 | 2.0 |
| LLaVA-NeXT-Video-7B (Liu et al., 2024a) | 32 | 21.8 | 25.6 | 6.2 |
| | 16 | 22.2 | 25.6 | 6.4 |
| | 8 | 21.8 | 25.6 | 6.4 |
| | 4 | 21.8 | 25.6 | 6.4 |
| | 2 | 21.2 | 25.4 | 6.0 |
| | 1 | 22.4 | 25.6 | 6.4 |
| Phi-3.5-Vision (Microsoft, 2024) | 32 | 22.0 | 21.2 | 4.8 |
| | 16 | 24.0 | 22.4 | 6.2 |
| | 8 | 21.8 | 21.2 | 5.0 |
| | 4 | 21.2 | 22.8 | 5.6 |
| | 2 | 20.4 | 21.6 | 3.8 |
| | 1 | 22.6 | 22.8 | 3.8 |

Table 8: Continuation of Table 7.

| Model | Frames | Text | Video | Group |
|---|---|---|---|---|
| MA-LMM-Vicuna-7B (He et al., 2024) | 32 | 22.4 | 25.6 | 6.8 |
| | 16 | 22.0 | 26.0 | 6.0 |
| | 8 | 23.0 | 26.0 | 6.4 |
| | 4 | 23.8 | 25.6 | 6.8 |
| | 2 | 23.8 | 25.6 | 6.8 |
| VideoLLaMA3 | 64 | **46.2** | **29.8** | 17.0 |
| | 32 | **47.4** | **29.2** | 15.0 |
| | 16 | **47.4** | **30.4** | 15.6 |
| | 8 | **43.4** | 28.6 | 12.0 |
| | 4 | **38.8** | 24.6 | 8.8 |
| | 2 | **35.6** | 22.8 | 7.4 |
| | 1 | 22.8 | 22.4 | 6.2 |
| LLaVA-Video-72B-Qwen2 | 128 | **45.2** | 28.8 | 16.4 |
| | 64 | **49.2** | **34.0** | **20.2** |
| | 32 | **48.4** | **33.2** | **20.0** |
| | 8 | **44.0** | 27.6 | 16.0 |
| | 4 | **37.2** | 23.0 | 10.2 |
| | 2 | **31.4** | 23.6 | 9.4 |
| | 1 | 25.2 | 26.8 | 8.0 |
| LLaVA-Video-7B-Qwen2 | 128 | **41.4** | 27.6 | 14.0 |
| | 64 | **42.4** | **30.0** | 17.0 |
| | 32 | **40.8** | **30.4** | 15.4 |
| | 16 | **36.8** | 28.0 | 13.0 |
| | 8 | **33.6** | 25.6 | 11.4 |
| | 4 | 29.0 | 24.6 | 10.0 |
| | 2 | 27.0 | 23.6 | 6.2 |
| | 1 | 27.8 | 22.4 | 6.4 |
| Aria | 32 | **34.8** | 28.8 | 12.0 |
| | 16 | **32.4** | 27.6 | 9.4 |
| InternVideo2.5-8B | 64 | **36.0** | 28.2 | 11.0 |
| | 32 | **35.0** | **29.0** | 11.4 |
| | 16 | **30.6** | 25.6 | 8.6 |
| | 8 | 23.4 | 25.0 | 6.0 |
| | 4 | 17.4 | 25.2 | 3.6 |
| Apollo-7B | 64 | **41.5** | **31.5** | 17.5 |
| | 32 | **44.3** | 28.5 | 15.8 |
| | 16 | **43.6** | 28.8 | 16.6 |
| | 8 | **42.6** | 28.4 | 14.2 |
| | 4 | **43.8** | **30.2** | 17.2 |
| InternVideo2.5-8B (Wang et al., 2025) | 32 | **35.0** | **29.0** | 11.4 |
| LLaVA-NeXT-7B (CoT) | 32 | 21.8 | 26.2 | 6.8 |
| LLaVA-NeXT-7B (Liu et al., 2024a) | 32 | 21.8 | 25.6 | 6.2 |
| $M^3$ (Cai et al., 2024) | 6 | 21.2 | 25.8 | 6.8 |
| Vicuna-7B-v1.5 (Zheng et al., 2023) | N/A | 22.2 | 25.6 | 6.4 |
| VTimeLLM (Huang et al., 2024) | 100 | 19.4 | 27.0 | 5.2 |
| LLaVA-OneVision-Qwen2-7B (Li et al., 2024a) | 64 | **40.2** | 28.6 | 12.6 |
| | 32 | **42.0** | 28.4 | 12.8 |
| | 16 | **41.6** | **29.4** | 14.6 |
| | 8 | **36.0** | 26.80 | 12.40 |
| | 4 | 29.2 | 28.0 | 10.0 |
| | 2 | 25.8 | 22.6 | 6.8 |

## L    VIDEO LENGTHS AND THE USE OF BLACK FRAMES

We report the video length distribution of our benchmark in Figure 11. We also report that out of the 1000 videos in Vinoground, there are a total of 992 videos with length $\leq$ 20 seconds, and 930 of them are $\leq$ 10 seconds.

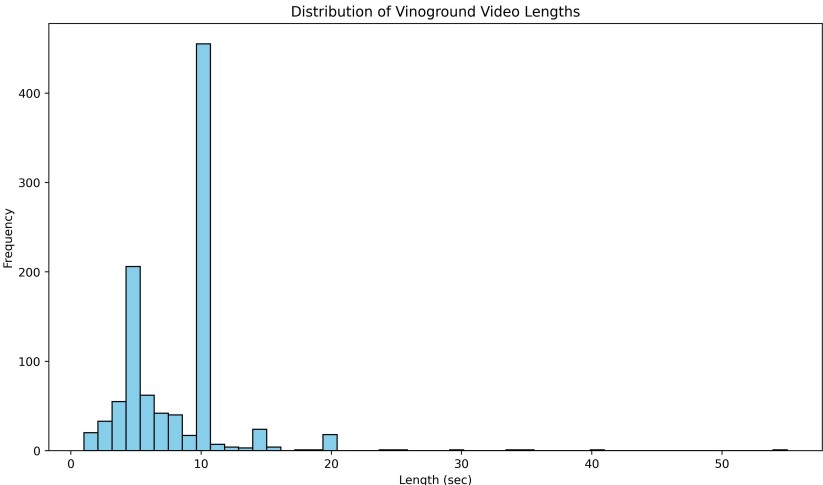

Figure 11: Video length distribution of Vinoground.

We show another histogram regarding—in all 500 concatenated videos for the video score metric— how much of each video is composed of black frames in Figure 12. We can see that for the majority, black frames only consist of less than two-tenths of the videos. This ensures that data loss due to sampling black frames is kept at a minimum.

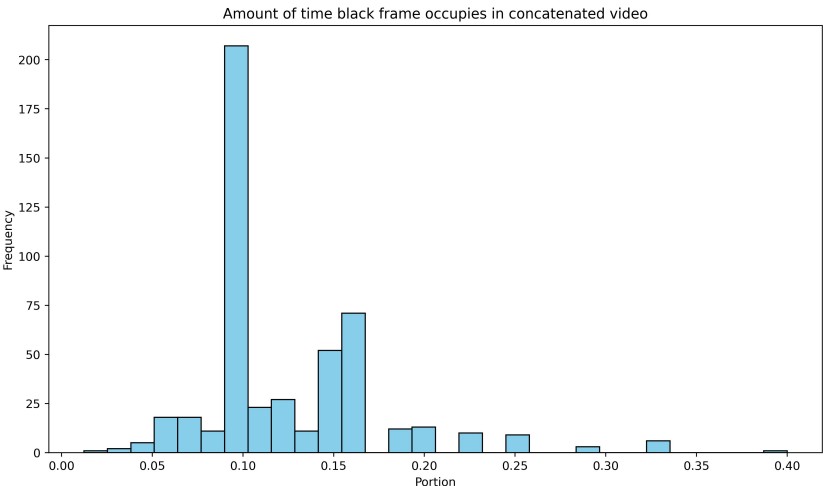

Figure 12: The portion of black frames in each concatenated video for video score questions.

# M    DETAILED CATEGORICAL TEASER

Figure 13: Examples of Vinoground video-caption pairs under each category.

## N    BENCHMARK DESIGN PHILOSOPHY AND ANALYSIS

This section provides further details on the design principles of Vinoground, discusses the implications of our task formulation, and offers insights for future research derived from our findings.

### N.1    ISOLATING TEMPORAL REASONING VIA COUNTERFACTUAL PAIRS

A core design principle of Vinoground is the use of counterfactual pairs to isolate temporal reasoning from static, atemporal cues. This addresses a common confound where models can achieve high performance on temporal tasks by relying on object recognition alone. For instance, without a counterfactual setup, a model tasked with "a person moves from left to right" might succeed simply by identifying `person` and `moves`, ignoring the directional dynamics.

Our approach, which we term "Looking Similar, Happening Differently," holds the core semantic elements (objects, actions, and caption vocabulary) constant across a pair while altering only the temporal sequence of events. In the same way that Singh et al. (2024) uses counterfactual audio to ensure true audiovisual synchronization, Vinoground uses counterfactual videos to ensure true visio-linguistic temporal reasoning. By presenting two distinct temporal realities for the same set of elements, our benchmark compels models to move beyond static analysis and engage with the dynamic progression of events over time.

### N.2    PROBING IMPLICIT TEMPORAL LOCALIZATION

While Vinoground is formatted as a multiple-choice task, its design inherently probes a model's ability to perform temporal localization. To correctly distinguish between "the cat moves before the person touches it" and "the person touches the cat before it moves," a model must implicitly localize the events "the cat moves" and "the person touches" within the video's timeline. Without the capacity to identify when these events occur relative to one another, a model cannot succeed.

This formulation demonstrates that even without explicit localization annotations, a significant gap exists between state-of-the-art models and the near-perfect human baseline. This highlights a fundamental weakness in the temporal reasoning capabilities of current video LLMs, regardless of the specific task format.

### N.3    A NOVEL CATEGORICAL ANALYSIS FOR THE TEMPORAL DOMAIN

Our primary contribution is the rigorous extension of fine-grained analysis to the temporal domain of video-language models. While the concept of models excelling at coarse-level tasks but failing at fine-grained details is known in image understanding, Vinoground provides the first comprehensive empirical evidence of this phenomenon in temporal reasoning, utilizing a benchmark of counterfactual, natural, and short-form videos.

Our design uniquely challenges models to discriminate between subtle temporal differences that other benchmarks might miss. This is underscored by the performance of leading models like GPT-4o, which achieves a 75.5% text score on the image-based Winoground benchmark but only 59.2% (with CoT) on Vinoground. This drop confirms that fine-grained temporal understanding in videos is substantially more challenging than in static images. Our analysis, therefore, offers novel insights into precisely how and where modern models fall short, guiding future development toward more nuanced temporal cognition.

### N.4    INSIGHTS FOR FUTURE MODEL DEVELOPMENT

The results from Vinoground offer valuable insights for the community. Our analysis in Table 3 reveals a significant performance gap between contrastive, CLIP-style models and text-generative Video LLMs, which we attribute to two main factors:

- **Richer Feature Representation:** Video LLMs utilize thousands of visual tokens, enabling a more detailed representation of temporal dynamics compared to the single-vector embeddings of CLIP-style models. The superior performance of Video-LLaVA over Lan-

guageBind, despite both using the same video encoder, suggests that the learning paradigm and architectural integration are critical.

- **Scale and Learning Objectives:** The larger scale and richer pretraining of the language models in Video LLMs likely contribute to their stronger baseline understanding of temporality. Furthermore, the shift from contrastive learning objectives to visually conditioned next-word prediction appears to better facilitate temporal understanding.

Based on these findings, we identify promising directions for future work:

- **Architectural Improvements:** Incorporating hierarchical temporal modeling or specialized cross-modal attention mechanisms could more effectively capture sequential dependencies and causality.
- **Data and Learning Strategies:** Developing pretraining datasets with more counterfactual examples could mitigate existing static biases. Fine-tuning LLMs on datasets that explicitly emphasize temporality remains a key avenue for improvement.

