# OpenReview forum: "Vinoground: Today’s LMMs Don’t Understand Short Counterfactual Videos"
_ICLR.cc/2026/Conference — ICLR 2026 Conference Withdrawn Submission_

### Official Review · Reviewer_eMTk · 2025-10-29

**Soundness:** 3
**Presentation:** 3
**Contribution:** 2
**Rating:** 4
**Confidence:** 4

**Summary:**

This paper argues that current large multimodal models are overestimated in their short video understanding capabilities, particularly in temporal reasoning. The authors propose Vinoground, a new benchmark consisting of 1,000 short video–caption counterfactual pairs designed to evaluate models’ ability to reason about event order and temporal relations. Experiments show that both proprietary and open-source models perform far below human accuracy, indicating that short video temporal understanding remains an unsolved challenge.

**Strengths:**

This paper presents a novel and well-designed benchmark for temporal counterfactual reasoning in short videos, offering a systematic test of large multimodal models’ true temporal understanding. Its originality lies in extending the Winoground concept to the temporal domain using natural short videos. The experiments are thorough, the evaluation metrics are carefully defined, and human baselines further validate task soundness. The writing is clear and well structured, with transparent data construction and evaluation procedures.

**Weaknesses:**

While the benchmark design is novel and well-motivated, the paper’s analytical depth is limited. The discussion mainly reports performance gaps without deeper insights into why models fail—such as the role of language priors, attention distribution, or deficiencies in temporal modeling. The dataset size (only 1,000 pairs) and limited diversity of scenes and actions constrain its representativeness and statistical robustness. Moreover, the paper does not propose any model-level or theoretical improvements, leaving its contribution primarily at the evaluation level.

**Questions:**

1) Is the scale and complexity of Vinoground sufficient to substantiate the strong claim that “current LMMs do not understand temporal relations”? With only 1,000 short clips, how do the authors ensure statistical reliability and representative coverage?

2) The observed performance gap is attributed to a lack of temporal reasoning, yet the paper does not clearly disentangle language bias, visual confusion, and temporal dependency failure. Can the authors provide ablation or control results validating that interpretation?

3) The paper asserts that LMMs have a “blind spot” in temporal understanding, but offers no mechanism-level analysis (e.g., cross-frame attention, video tokenization). How can the authors be sure the failure arises from temporal reasoning itself rather than other vision–language alignment issues?

4) Given that the dataset mainly consists of simple, short human actions, would the same conclusions hold in longer or multi-agent scenarios with causal dependencies? Do the authors plan to test this assumption?

---

### Official Review · Reviewer_xsgh · 2025-10-31

**Soundness:** 1
**Presentation:** 2
**Contribution:** 1
**Rating:** 2
**Confidence:** 4

**Summary:**

The manuscript introduces Vinoground, a new benchmark to evaluate large multimodal models (LMMs) on temporal counterfactual reasoning in short videos (less than 10 seconds). It includes 1000 natural video-caption pairs, each with counterfactual captions differing only by temporal order. Experiments show that all models, including GPT-4.1, Gemini-2.5-Pro, and InternVL3-78B, perform far below humans, with the best model (OpenAI’s o3) achieving only 56.6% group score versus 90% for humans. The results reveal that current LMMs fail at dense temporal reasoning, even on short clips, exposing a key gap in multimodal understanding.

**Strengths:**

1, Introduces a benchmark specifically designed for temporal counterfactual reasoning, addressing a  gap in current evaluation.

2, The dataset design uses GPT-4 for controlled caption generation and human validation for quality assurance.

3, Providesquantitative analysis across model families, metrics, and video categories, highlighting consistent failure patterns.

**Weaknesses:**

1. The dataset heavily relies on existing VATEX videos, limiting diversity and originality of visual content.

2. Manual curation and subjective filtering introduce potential annotation bias, yet no inter-rater reliability or statistical validation is reported.

3. The paper lacks model ablations or diagnostic experiments to identify why models fail; whether due to architecture, pretraining, or tokenization limits.

4. Authors should go beyhond this preliminary study by developing:  (i) controlled generation of synthetic but naturalistic videos to expand coverage, (ii) quantitative validation of human curation quality, and (iii) architectural or attention-map analyses explaining model failure mechanisms.

5. I suggest integrating temporal-attention fine-tuning baselines to strengthen the empirical contribution.

6. Overall, its scientific novelty and analytical depth are limited. The benchmark construction is incremental, mainly extending Winoground to short videos—and the experimental section is descriptive rather than diagnostic. There is no real modeling insight or methodological innovation, and the reliance on off-the-shelf models weakens originality.

7. In its current form, the work reads as a dataset release without substantive conceptual contribution and is not ready for acceptance at ICLR.

**Questions:**

Please see weakness.

**Details Of Ethics Concerns:**

-

---

### Official Review · Reviewer_ziq2 · 2025-11-02

**Soundness:** 2
**Presentation:** 2
**Contribution:** 2
**Rating:** 2
**Confidence:** 4

**Summary:**

This paper introduces Vinoground, a temporal counterfactual short video LMM evaluation benchmark, inspired by Winoground. The authors note that the benchmark differentiates itself from existing ones by focusing on short, natural counterfactual videos. The benchmark consists of 3 major categories, “object”, “action”, and “viewpoint”, and 4 minor categories, “interaction”, “cyclical”, “spatial”, and “contextual”. The evaluation uses text score, video score, and group score to evaluate the models’ ability to match caption to video, video to caption, and combined, demonstrating LMMs' limitation in temporal reasoning.

**Strengths:**

1. The benchmark highlights an important limitation of LMMs in short video comprehension.
2. The authors have done extensive experiments on different models using the 3 different evaluation metrics.
3. The authors provide analysis of model performance based on model types (CLIP-based, text-generative, reasoning).

**Weaknesses:**

1. The motivation behind this work is a known problem in the field and this benchmark contributes limited novelty to the field as short counterfactual videos are studied across existing benchmarks (e.g. action/event order is a well-established category in prior work). As the authors highlight the importance of “natural” video, it remains unclear how these curated natural videos impose stricter temporal reasoning constraints on the models compared to edited videos from existing benchmarks (e.g. [1], [2], [3]) without experimental evidence.
2. The use of GPT-4 creates a “chicken and egg problem”. As acknowledged by existing works (e.g. [1], [2], [3]) that current models lack the capability to reason temporally, using the model to create temporal counterfactuals does not guarantee the temporal constraint without additional supporting evidence.
3. It remains unclear why the caption generation process comes before the video curation and how they are made sure to be closely aligned.
4. There are claims made in the paper that could be untrue/controversial. I suggest the authors be cautious with making such claims without providing supporting evidence.
    - L50: “This has led many researchers to believe that short video comprehension has mostly been solved…” The emergence of many works (e.g. [1], [2], [3], [4], [5], etc) clearly shows the opposite.
    - L93: “Vinoground, the first temporal and natural counterfactual evaluation benchmark for evaluating video understanding models using only short videos.” [1], [2] should already contain natural temporal counterfactual short videos.
    - L177: “making it the most novel temporal reasoning benchmark.” It is a controversial claim considering my comment in 1.


[1] Li, Shicheng, et al. "Vitatecs: A diagnostic dataset for temporal concept understanding of video-language models." European Conference on Computer Vision. Cham: Springer Nature Switzerland, 2024.\
[2] Liu, Yuanxin, et al. "Tempcompass: Do video llms really understand videos?." arXiv preprint arXiv:2403.00476 (2024).\
[3] Shangguan, Ziyao, et al. "Tomato: Assessing visual temporal reasoning capabilities in multimodal foundation models." arXiv preprint arXiv:2410.23266 (2024).\
[4] Cai, Mu, et al. "Temporalbench: Benchmarking fine-grained temporal understanding for multimodal video models." arXiv preprint arXiv:2410.10818 (2024).\
[5] Cores, Daniel, et al. "Lost in Time: A New Temporal Benchmark for VideoLLMs." arXiv preprint arXiv:2410.07752 (2024).

**Questions:**

1. Why are the caption pairs generated before looking at any video references? Have the authors considered locating at least one video and then generating the caption pairs to ensure that the caption closely aligns with the content of the video? How are the authors making sure that the video pairs curated exactly match the caption contents? Are there any experiment results?
2. If video pairs curated from VATEX are two separate clips that match the positive-negative caption pairs, how are they “counterfactual” as they are separate natural scenes that happened in real life?
3. As the authors have brought up several times the notion of “fine-grained details” (e.g. “missing fine-grained details”, “evaluate LMMs in a fine-grained manner”, and “how fine-grained comprehension is also crucial for dense temporal reasoning”), how is this notion related to the claims made about this benchmark and to the central argument of this paper?

**Details Of Ethics Concerns:**

The authors curate videos from YouTube but do not clarify how copyright and redistribution are handled in their benchmark.

---

### Official Review · Reviewer_jeu5 · 2025-11-02

**Soundness:** 3
**Presentation:** 2
**Contribution:** 2
**Rating:** 4
**Confidence:** 3

**Summary:**

The authors propose to study whether current large multimodal models entail adequate reasoning capabilities for even short duration videos (in comparison to growing focus on long form videos) with counterfactual caption pairs . Accordingly, authors introduce a benchmark named Vinoground (similar in spirit to the Winoground benchmark) which comprises 500 counterfactual text pairs with corresponding videos of max 10 seconds. GPT is used to generate counterfactual captions of different categories, and matching videos are then obtained from VATEX dataset and similarity matching. Human annotators (the authors themselves) perform quality control to check if caption is a good match for the video. For evaluation, a 'text score' and 'video score' to test temporal reasoning capabilities across different input forms, and a cumulative 'group score' metric.

**Strengths:**

1. The paper studies an interesting problem on whether modern LMMs are capable at temporal reasoning capabilities within even short videos (where models can't suffer from long context capabilities)
2. The methodology regarding dataset construction and evaluation is mostly adequately described.
3. Results show that models perform fairly poorly on the benchmark and lag behind humans on the composite score. A wide number of models are considered and human performances are also noted for comparison.

**Weaknesses:**

1. There have been multiple previous works that have explored creation of temporal counterfactual question-video pairs to assess basic reasoning capabilities of models. These include "Test of Time: Instilling Video-Language Models with a Sense of Time" [1] and CLAVI (Complements in Language and Video) [2] which have largely done similar work in creating complementary/counterfactual pairs of text and video, and demonstrating how video-language models show inherent biases and weak performances on simple temporal reasoning scenarios. These works are not mentioned and should be differentiated from. Currently, the novelty of this work seems to be limited and it seems an extension of these works evaluated for more recent LMM models.

2. A more detailed analysis into limitations of current models on constructed benchmark can be useful. This could help differentiate the work further from previous similar benchmarks, authors could consider doing a more detailed error analysis and study into how different components of open source models (such as Qwen or InternVL) impact the model's reasoning ability. E.g. some questions to explore could be:
- do limitations or biases arise from the backbone LLM of these models or is it the visual backbone model?
- does the way attention is performed b/w image and video inputs (e.g. cross attention vs concatenated attention) influence results?
- does chain-of-thought or other prompting strategies impact performances, and is produced chain of thought sensical/reasonable?

(To make space for point 2, less important details on experimental setup and dataset construction can be moved to appendix)


[1] Bagad, Piyush, Makarand Tapaswi, and Cees GM Snoek. "Test of time: Instilling video-language models with a sense of time." Proceedings of the IEEE/CVF Conference on Computer Vision and Pattern Recognition. 2023.

[2] Rawal, Ishaan Singh, et al. "Dissecting Multimodality in VideoQA Transformer Models by Impairing Modality Fusion." International Conference on Machine Learning. PMLR, 2024.

**Questions:**

Please see weaknesses.

---

### Note · Authors · 2025-11-13

I have read and agree with the venue's withdrawal policy on behalf of myself and my co-authors.